# Meta Prompting: A Framework for Agentic and Compositional Reasoning

## Abstract

We introduce Meta Prompting (MP), a framework that elevates the reasoning capabilities of large language models (LLMs) by focusing on the formal structure of a task rather than content-specific examples. We establish a theoretical foundation for this paradigm, formalizing MP as a functor that maps a category of tasks to a category of structured prompts, thereby guaranteeing that compositional problem-solving strategies can be systematically decomposed into modular prompt structures. We extend this concept to Recursive Meta Prompting (RMP), an automated process where an LLM can generate and refine its own prompts. We model this self-improvement loop formally as a monad, providing a principled framework for automated prompt engineering. Our claims are validated through several experiments demonstrating that a Qwen-72B base model, guided by a single, example-agnostic meta-prompt, achieves improved results on MATH, GSM8K, and Game of 24. These results are achieved with substantial token efficiency gains over traditional few-shot methods.

## 1 Introduction

The advent of foundation models, particularly Large Language Models (LLMs), has transformed the research of various fields. With extensive training data and robust generalization capabilities, these models have significantly broadened the horizons of computational linguistics, text understanding, and text generation (Devlin et al., 2018; Radford et al., 2018; 2019; Brown et al., 2020; Raffel et al., 2020; OpenAI, 2023). Despite these advances, LLMs still exhibit limitations when addressing complex reasoning tasks—especially those demanding deep, abstract thought such as advanced mathematics (Lightman et al., 2023). This observation underscores the need for methodologies that enhance the reasoning faculties of LLMs.

A core challenge originates from the auto-regressive token prediction architecture that underpins modern LLMs (Radford et al., 2018; 2019; Brown et al., 2020). While this design excels at a broad range of tasks, it is not inherently configured for the depth and sophistication of human-like analytical reasoning. This discrepancy is aptly captured by the dual-process theory of cognitive psychology (Kahneman, 2011), which distinguishes between the rapid, intuitive responses of System 1 and the deliberate, systematic processes of System 2. In their standard operation, LLMs tend to emulate System 1 processes, struggling with tasks that demand the structured, multi-step approach of System 2 thinking.

In recent years, approaches such as Chain-of-Thought (CoT) (Wei et al., 2022) and Tree-of-Thought (ToT) (Yao et al., 2023; Long, 2023) prompting have been proposed to guide LLMs toward more deliberative reasoning. While these methods have improved performance, they primarily rely on content-based examples and lack a formal, compositional structure for building complex reasoning processes.

In response to these challenges, we introduce Meta Prompting (MP), a paradigm that shifts the focus from content-based analogy to formal procedural guidance. Instead of providing examples of what to think, a meta-prompt provides a structured template for how to think. We establish a theoretical foundation for MP, formalizing it as a functorial mapping from a category of tasks to a category of prompts. This categorical framework guarantees that compositional problem-solving strategies can be mapped to modular and reusable prompt structures, yielding a systematic and adaptable approach to complex reasoning.

Integrate step-by-step reasoning to solve mathematical problems under the following structure:
{
    "Problem": "[question to be answered]",
    "Solution": {
        "Step 1": "Begin the response with 'Let's think step by step.' ",
        "Step 2": "Follow with the reasoning steps, ensuring the solution process is broken down clearly and logically.".
        "Step 3": "End the solution with the final answer encapsulated in a LaTeX-formatted box, ⬚, for clarity and emphasis."
    },
    "Final Answer": "[final answer to the problem]"
}

Figure 1: A structure meta prompt presented in JSON format.

A pivotal innovation of our work is the application of this framework recursively, a concept we term Recursive Meta Prompting (RMP). Analogous to metaprogramming, RMP enables an LLM to autonomously generate and refine its own prompts. We model this self-improvement process formally using a monad, providing a principled framework for a model to not only solve problems, but to learn how to improve its own problem-solving strategies. This self-referential capability is a significant step towards greater model autonomy and automated prompt engineering.

The efficacy of our framework is validated through extensive experiments on challenging benchmarks, including the Game of 24 (Yao et al., 2023), GSM8K (Cobbe et al., 2021), and MATH (Hendrycks et al., 2021).

In summary, our contributions are as follows:

- We introduce Meta Prompting (MP) and formalize it using category theory. We model MP as a functor that preserves compositional structure, and we prove a proposition on its compositional properties. We further introduce Recursive Meta Prompting (RMP) and model it with a monadic framework that provides a principled basis for prompt self-improvement.
- Through proof-of-concept experiments, we show that a Qwen-72B *base* language model equipped with Meta Prompting—without additional instruction tuning—achieves a PASS@1 accuracy of 46.3% on MATH, 83.5% on GSM8K, and a 100% success rate on Game of 24, competitive with prior reports under CoT for early GPT-4 (2023–0314) while using an example-free, structure-only prompt.

**Problem Statement**:

- **Problem**: [question to be answered]

**Solution Structure**:

1. Begin the response with "Let's think step by step."
2. Follow with the reasoning steps, ensuring the solution process is broken down clearly and logically.
3. End the solution with the final answer encapsulated in a LaTeX-formatted box, ⬚, for clarity and emphasis.
4. Finally, state "The answer is [final answer to the problem].", with the final answer presented in LaTeX notation. ———

Figure 2: A structured meta prompt presented in markdown format for solving MATH (Hendrycks et al., 2021) problems, as introduced in the Minerva study by (Lewkowycz et al., 2022).

Problem: Find the domain of the expression $\frac{\sqrt{x-2}}{\sqrt{5-x}}$.
Solution: The expressions inside each square root must be non-negative. Therefore, $x - 2 \geq 0$, so $x \geq 2$, and $5 - x \geq 0$, so $x \leq 5$. Also, the denominator cannot be equal to zero, so $5 - x > 0$, which gives $x < 5$. Therefore, the domain of the expression is $\boxed{[2, 5)}$. Final Answer: The final answer is $[2, 5)$. I hope it is correct.

———

Problem: If $\det \mathbf{A} = 2$ and $\det \mathbf{B} = 12$, then find $\det(\mathbf{AB})$.
Solution: We have that $\det(\mathbf{AB}) = (\det \mathbf{A})(\det \mathbf{B}) = (2)(12) = \boxed{24}$. Final Answer: The final answer is 24. I hope it is correct.

———

...

Figure 3: An example of the most widely used few-shot prompt for solving MATH problems. **Note:** In contrast, our meta-prompt in Fig. 2 is *generated via RMP* from a single task-agnostic meta-meta-prompt (Sec. 4).

## 2 Background

Category theory provides a high-level language for describing mathematical structures and their relationships. We use it to formalize the relationship between task structures and prompt structures.

### 2.1 Category Theory

**Definition 2.1** (Category). A *category* $\mathscr{C}$ comprises a collection of *objects* and, for each pair of objects $A, B \in \mathscr{C}$, a set of *morphisms* (or arrows) from $A$ to $B$, denoted as $\mathrm{Hom}(A, B)$. Morphisms can be intuitively understood as directed connections or mappings between objects. Notably, in a locally small category, morphisms between any two objects form a set, rather than a class.

**Definition 2.2** (Morphisms). For objects $A, B$ in a category $\mathscr{C}$, a morphism $f$ from $A$ to $B$ is denoted by $f : A \to B$, where $A$ is the source, and $B$ is the target. It is assumed that $\mathrm{Hom}(A, B)$ is disjoint from $\mathrm{Hom}(A', B')$ unless $A = A'$ and $B = B'$.

**Definition 2.3** (Composition of Morphisms). Morphisms in a category are composed in an associative manner. Specifically, if $f \in \mathrm{Hom}(A, B)$ and $g \in \mathrm{Hom}(B, C)$, their composition is a morphism $g \circ f \in \mathrm{Hom}(A, C)$. This composition obeys the associative law: given $f \in \mathrm{Hom}(A, B)$, $g \in \mathrm{Hom}(B, C)$, and $h \in \mathrm{Hom}(C, D)$, it holds that $h \circ (g \circ f) = (h \circ g) \circ f$.

**Definition 2.4** (Identity Morphisms). Each object $A$ in a category $\mathscr{C}$ possesses an *identity morphism* $\mathrm{id}_A : A \to A$. This morphism, when composed with any other morphism $f : A \to B$ or $g : B \to A$, yields the original morphism: $f \circ \mathrm{id}_A = f$ and $\mathrm{id}_B \circ g = g$. Furthermore, identity morphisms are unique to each object.

### 2.2 Functors

**Definition 2.5** (Covariant Functor). A *covariant functor* F from a category $\mathscr{A}$ to a category $\mathscr{B}$, denoted $\mathrm{F} : \mathscr{A} \to \mathscr{B}$, consists of two key components:

- A mapping of objects: $\mathrm{F} : \mathrm{obj}(\mathscr{A}) \to \mathrm{obj}(\mathscr{B})$.
- For each pair of objects $A_1, A_2 \in \mathscr{A}$ and a morphism $m : A_1 \to A_2$, a corresponding morphism $\mathrm{F}(m) : \mathrm{F}(A_1) \to \mathrm{F}(A_2)$ in $\mathscr{B}$.

This functor respects both identity morphisms ($\mathrm{F}(\mathrm{id}_A) = \mathrm{id}_{\mathrm{F}(A)}$) and composition ($F(m_2 \circ m_1) = F(m_2) \circ F(m_1)$).

**Definition 2.6** (Contravariant Functor). A *contravariant functor* is similar to a covariant functor, but it reverses the direction of the morphisms: for $m : A_1 \to A_2$, the functor maps it to a morphism from $F(A_2)$ to $F(A_1)$. Formally, $F(m_2 \circ m_1) = F(m_1) \circ F(m_2)$.

### 2.3 Natural Transformations

**Definition 2.7** (Natural Transformation). A *natural transformation* between two covariant functors $F, G : \mathscr{A} \to \mathscr{B}$ is a family of morphisms $\{m_A : F(A) \to G(A)\}_{A \in \mathscr{A}}$ such that for every morphism $f : A \to A'$ in $\mathscr{A}$, the corresponding diagram commutes. When each $m_A$ is an isomorphism, the transformation is a *natural isomorphism.*

This concept is crucial for understanding the RMP monad, where the unit ($\eta$) and multiplication ($\mu$) are natural transformations.

### 2.4 Monads in Category Theory

**Definition 2.8** (Monad). A *monad* on a category $\mathscr{C}$ is a triple $(\mathrm{T}, \eta, \mu)$ consisting of:
  item An *endofunctor* $\mathrm{T} : \mathscr{C} \to \mathscr{C}$.
- A natural transformation $\eta : \mathrm{Id}_{\mathscr{C}} \to \mathrm{T}$, called the *unit* (or return).
- A natural transformation $\mu : \mathrm{T} \circ \mathrm{T} \to \mathrm{T}$, called the *multiplication* (or join).

These components must satisfy coherence conditions known as the monad laws (associativity and left/right identity), which ensure that compositions of monadic operations behave in a well-structured manner. In computer science, monads are fundamental for modeling computations with side effects, such as state, I/O, or, as we will argue, recursive prompt refinement.

## 3 Meta Prompting

Meta Prompting is a prompting technique that emphasizes the structural and syntactical aspects of problems by prioritizing the overall format and pattern over specific content details. This method constructs an abstract and structured approach to interacting with large language models (LLMs), placing emphasis on the form and syntax of information. Such an approach is particularly effective in scenarios where recognizing the underlying framework of a problem is crucial for its resolution.

**Definition 3.1** (Meta Prompt). A *Meta Prompt* is an example-agnostic structured prompt designed to capture the reasoning structure of a specific category of tasks. It provides a scaffold that outlines the general approach to a problem, thereby enabling LLMs to fill in task-specific details as needed. This methodology focuses on the procedural aspects of problem-solving, the *how*, rather than the content, specific details, the *what.*

This emphasis on structure is analogous to type theory (see Appendix A.1), where each component of a prompt can be assigned a "type" (e.g., 'ProblemStatement: string', 'ReasoningStep: list[string]', 'FinalAnswer: float'). A meta-prompt thus defines a "type signature" for the desired output, guiding the LLM to generate responses that are not only semantically relevant but also syntactically correct according to the specified format. Examples of such structured prompts are shown in Figures 1 and 2.

### 3.1 Formalizing Meta Prompting

In category theory, a functor $F$ from a category $\mathcal{C}$ to a category $\mathcal{D}$, denoted $F : \mathcal{C} \to \mathcal{D}$, maps objects and morphisms (arrows) from $\mathcal{C}$ to $\mathcal{D}$ in a manner that preserves the categorical structure (i.e., identity morphisms and composition of morphisms).

Applying this concept to Meta Prompting, we define two categories:

**Definition 3.2** (Categories of Tasks and Prompts). Let $\mathcal{T}$ denote a category whose objects are various tasks or problems (e.g., mathematical problems, coding challenges, or theoretical queries). The morphisms in $\mathcal{T}$, denoted $\mathrm{Hom}_{\mathcal{T}}(X, Y)$, represent the methodologies or transformations that relate one problem $X$ to another $Y$ (for instance, transforming a linear algebra problem into an optimization problem).

Similarly, let $\mathcal{P}$ denote a category whose objects are structured prompts designed to guide the solution of these tasks. Objects in $\mathcal{P}$ comprise carefully crafted prompts—such as a step-by-step guide for solving a

differential equation or a template for writing code—while the morphisms, denoted $\mathrm{Hom}_{\mathcal{P}}(U, V)$, represent the adaptations or refinements of one prompt $U$ into another $V$ (e.g., adapting a prompt for a basic algebra problem to one suited for a complex calculus problem).

The core of our framework is the *Meta Prompting Functor*:

**Definition 3.3** (Meta Prompting Functor)**.** Define the *Meta Prompting Functor* $\mathcal{M} : \mathcal{T} \to \mathcal{P}$ as follows:

- **On Objects:** For each task $X \in \mathcal{T}$, assign a corresponding structured prompt $\mathcal{M}(X) \in \mathcal{P}$. For example, if $X$ is a quadratic equation problem, then $\mathcal{M}(X)$ may be a prompt outlining the necessary steps to solve quadratic equations.
- **On Morphisms:** For each morphism $f : X \to Y$ in $\mathcal{T}$, which represents a transformation or method for solving task $X$ in terms of task $Y$, assign a morphism $\mathcal{M}(f) : \mathcal{M}(X) \to \mathcal{M}(Y)$ in $\mathcal{P}$. For instance, if $f$ transforms a basic algebra task into an advanced algebra problem, then $\mathcal{M}(f)$ adapts the corresponding prompt accordingly.

A functor must preserve the categorical structure; that is, for any $f : X \to Y$ and $g : Y \to Z$ in $\mathcal{T}$, $\mathcal{M}(g \circ f) = \mathcal{M}(g) \circ \mathcal{M}(f)$, and for every object $X \in \mathcal{T}$, $\mathcal{M}(\mathrm{id}_X) = \mathrm{id}_{\mathcal{M}(X)}$.

Meta Prompting thus provides a systematic method for constructing prompts tailored to specific task categories. This approach ensures that a language model equipped with the appropriate prompt accurately captures the task's objectives and executes the solution process as intended. Its adaptability further allows for effective application even when the task category is not naturally representable in the language (textual, visual, or programming) of the model.

The preservation of composition, $\mathcal{M}(g \circ f) = \mathcal{M}(g) \circ \mathcal{M}(f)$, is not merely a mathematical formality; it is the theoretical guarantee of modularity and systematic problem decomposition. It implies that if a complex problem-solving strategy can be constructed by composing simpler strategies, the corresponding prompt can also be constructed by composing simpler prompts in a principled manner. This insight forms the basis for building complex reasoning chains from fundamental building blocks. We formalize this as a proposition.

**Compiler view.** Practically, objects (tasks) correspond to typed *schemas* for prompts (e.g., `ProblemStatement`, `Steps`, `FinalAnswer`); morphisms are *schema-preserving edits* (e.g., adding a "compute" section after "parse"). Functoriality ensures that a task reduction like *parse $\to$ compute* compiles to a modular prompt whose sections compose in the same order.

**Theorem 3.4** (Compositionality of Meta Prompting)**.** Let a task $T \in \mathcal{T}$ be composed of sub-tasks $T_1$ and $T_2$ via a composition of morphisms, such that $T$ is the result of applying transformation $f : T_1 \to T_2$ followed by $g : T_2 \to T_3$. The meta-prompt for the composite task, $\mathcal{M}(g \circ f)$, is equivalent to the composition of the meta-prompts for the sub-tasks, i.e., $\mathcal{M}(g) \circ \mathcal{M}(f)$.

*Proof Sketch.* The theorem holds by the definition of $\mathcal{M}$ as a functor. A functor is a structure-preserving map between categories, which by definition must preserve the composition of morphisms. Therefore, the compositional structure of tasks in $\mathcal{T}$ is necessarily preserved in the structure of prompts in $\mathcal{P}$. This property ensures that breaking down a complex problem into a sequence of simpler steps in the task domain corresponds directly to a sequence of prompt transformations in the prompt domain. $\square$

This mapping can be hand-crafted by a human or generated using LLMs via a recursive, self-composing method (see Section 4). Furthermore, morphisms in $\mathcal{T}$ (representing transformations between tasks) are mapped to corresponding morphisms in $\mathcal{P}$ (representing transformations between prompts) in such a way that the structure and logic of problem-solving are preserved.

**Example of Meta Prompting.** Consider the task of solving a quadratic equation, represented as an object $Q \in \mathcal{T}$. The Meta Prompting functor $\mathcal{M}$ maps $Q$ to a structured prompt $P \in \mathcal{P}$ (see Figure 4), which might outline steps such as identifying coefficients, computing the discriminant, and applying the quadratic formula. This structured prompt guides the LLM in solving any quadratic equation by following a systematic approach, regardless of the specific coefficients involved.

A structure meta prompt for solving quadratic equations in the form $ax^2 + bx + c = 0$:
{
    "Problem": "Solve the quadratic equation $ax^2 + bx + c = 0$ for $x$.",
    "Solution": {
        "Step 1": "Identify the coefficients $a$, $b$, and $c$ from the equation.",
        "Step 2": "Compute the discriminant using $\Delta = b^2 - 4ac$.",
        "Step 3": "Determine the nature of the roots by checking if $\Delta > 0$, $\Delta = 0$, or $\Delta < 0$.",
        "Step 4": "If $\Delta > 0$, calculate the two distinct real roots using $x_{1,2} = \frac{-b \pm \sqrt{\Delta}}{2a}$.",
        "Step 5": "If $\Delta = 0$, calculate the single real root using $x = \frac{-b}{2a}$.",
        "Step 6": "If $\Delta < 0$, calculate the complex roots using $x_{1,2} = \frac{-b \pm i\sqrt{|\Delta|}}{2a}$.",
        "Step 7": "Conclude by summarizing the roots in a LaTeX-formatted box, $\boxed{...}$."
    },
    "Final Answer": "Depending on the value of $\Delta$, the final answer is provided by $x_{1,2}$."
}

Figure 4: A structured meta prompt in JSON format for solving quadratic equations.

Example 1:
Problem: Solve the quadratic equation $3x^2 + 4x - 5 = 0$.
Solution:

- Step 1: Identify the coefficients: $a = 3$, $b = 4$, and $c = -5$.
- Step 2: Calculate the discriminant: $\Delta = b^2 - 4ac = 4^2 - 4(3)(-5) = 16 + 60 = 76$.
- Step 3: Since $\Delta > 0$, the equation has two distinct real roots.
- Step 4: Compute the roots using the quadratic formula: $x_{1,2} = \frac{-b \pm \sqrt{\Delta}}{2a} = \frac{-4 \pm \sqrt{76}}{6}$.
- Step 5: Simplify to obtain the roots: $x_1 = \frac{-4 + \sqrt{76}}{6}$ and $x_2 = \frac{-4 - \sqrt{76}}{6}$.

Final Answer: The roots of the equation $3x^2 + 4x - 5 = 0$ are $x_1 = \frac{-4 + \sqrt{76}}{6}$ and $x_2 = \frac{-4 - \sqrt{76}}{6}$.

Example 2: ...

Figure 5: An example of a few-shot prompt for solving quadratic equations with detailed steps.

**Characteristics of Meta Prompting.** Meta Prompting prioritizes form and structure over content by employing a syntactical template that guides the expected response or solution. It utilizes abstract examples to illustrate the overall structure of problems and solutions, without focusing on specific details. Drawing from type theory, Meta Prompting emphasizes categorizing prompt components, such as problem statements, solution steps, and conclusions, and arranging them logically to ensure a coherent problem-solving process. This versatile technique is applicable across various domains, offering a clear roadmap for navigating complex topics.

### 3.2 Distinctions between Meta Prompting and Few-Shot Prompting

Meta Prompting differs from Few-Shot Prompting in both its methodology and objectives. Few-shot prompting provides a limited set of concrete, content-rich '(problem, solution)' pairs to guide the model via in-context analogy. In contrast, Meta Prompting provides a single, content-agnostic structural template that outlines the reasoning process. It teaches the model how to think, whereas few-shot prompting shows the model what has been thought.

Beyond few-shot methods, Meta Prompting also distinguishes itself from other programmatic or structured prompting approaches, such as those using XML tags or frameworks like DSPy (Khattab et al., 2023). While these methods also impose structure, they often function as programming layers that compile into traditional few-shot or zero-shot prompts. Meta Prompting, as formalized here, is a more fundamental concept focused on the direct, example-agnostic mapping between a task's abstract structure and a prompt's syntactic structure. For further illustration of these differences, please refer to Figures 1, 2, and 3.

### 3.3 Meta Prompting for Complex Reasoning

Integrating Meta Prompting within AI systems enhances their capability to interact with symbolic systems and code environments. By utilizing typed, structured prompts, AI models can more effectively parse and interpret symbolic information, a crucial advantage in fields such as mathematics and logic. Moreover, the structured nature of these prompts aligns seamlessly with code environments, enabling AI agents to understand, modify, and execute code across both textual and visual programming paradigms. This broadened interaction fosters a more comprehensive understanding across various domains. This specialized application of Meta Prompting for complex reasoning is tailored to address intricate, multi-layered problems that demand profound analytical and logical processing. (For an illustrative example, see Figure 9 in Appendix B.)

### 3.4 Advantages of Meta Prompting

Meta Prompting offers distinct benefits over traditional few-shot approaches, particularly for large language models (LLMs). Two primary advantages are:

**Token Efficiency.** By emphasizing structure over exhaustive content, Meta Prompting significantly reduces the number of tokens required. This efficiency is vital in contexts where token limits are imposed. The focus on syntax ensures a concise yet clear representation of problems.

**Fair Comparison and Zero-Shot Efficacy.** Meta Prompting can be regarded as a form of zero-shot prompting, wherein the influence of specific examples is minimized (Brown et al., 2020; Liu et al., 2020; Reynolds & McDonell, 2021). This approach enables a more equitable comparison among different problem-solving models by avoiding reliance on example-based learning and specific prior knowledge. Consequently, the LLM can approach problems with a fresh, unbiased perspective, unencumbered by the limitations of few-shot examples.

In summary, Meta Prompting is distinguished by its token efficiency and its ability to provide a fair, unbiased approach to problem-solving, making it especially valuable in settings where token economy and equitable model comparisons are critical.

---

**Algorithm 1** Recursive Meta Prompting (RMP)

---
1: **Input:** Initial task description $T_0$, Meta-Meta-Prompt $P_{\text{meta}}$, LLM $\mathcal{L}$
2: $P_{\text{current}} \leftarrow \text{InitialPrompt}(T_0)$ {Generate a basic prompt}
3: **for** $i = 1$ **to** $N_{\text{max\_iterations}}$ **do**
4:     $P_{\text{refined}} \leftarrow \mathcal{L}(P_{\text{meta}}, P_{\text{current}})$ {Refine the prompt}
5:     **if** $\text{IsConverged}(P_{\text{refined}}, P_{\text{current}})$ **then**
6:         **break**
7:     **end if**
8:     $P_{\text{current}} \leftarrow P_{\text{refined}}$
9: **end for**
10: $Solution \leftarrow \mathcal{L}(P_{\text{current}}, T_0)$ {Solve task with the final prompt}
11: **return** $Solution$

---

## 4 Recursive Meta Prompting: Self-Refinement and Automation

While Meta Prompting provides a robust framework for solving external tasks, its most powerful application lies in turning the lens of prompting inward: using meta-prompts to generate and refine other prompts. We

call this process **Recursive Meta Prompting (RMP)**. RMP endows an AI system with the capacity for autonomous self-improvement, where the model not only executes tasks but also actively constructs and enhances its own guiding instructions. This paradigm mirrors the concept of metaprogramming in computer science, where a program can treat other programs (or itself) as data to be analyzed and modified.

### 4.1 A Monadic Framework for Prompt Refinement

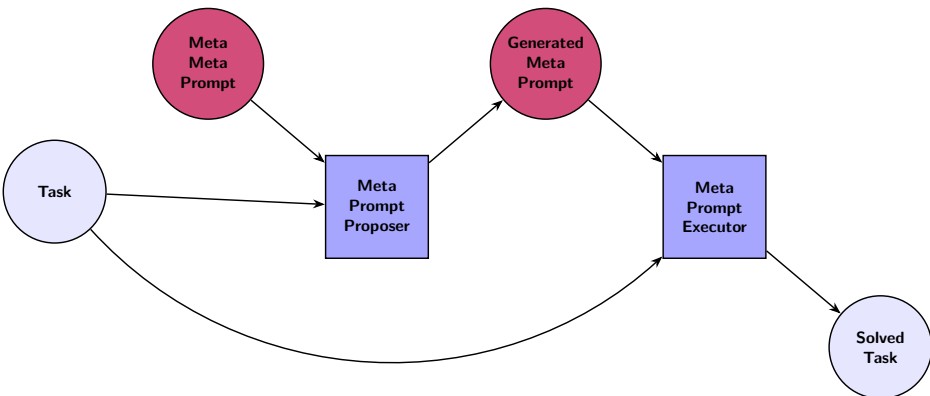

Figure 6: The workflow of Recursive Meta Prompting. A *Meta-Meta-Prompt* guides a Proposer LLM to generate a task-specific Meta Prompt. This generated prompt is then used by an Executor LLM to solve the original task.

The process of recursive refinement is elegantly captured by the mathematical structure of a monad. We can model RMP as an endofunctor on the category of prompts, $\mathcal{M}_\mathcal{P} : \mathcal{P} \to \mathcal{P}$, which takes a prompt and outputs a refined version. This forms a monad $(\mathcal{M}_\mathcal{P}, \eta, \mu)$:

- **Endofunctor** $\mathcal{M}_\mathcal{P}$: The core refinement operation. Given a prompt $P$, $\mathcal{M}_\mathcal{P}(P)$ is the improved prompt.
- **Unit** ($\eta : \mathrm{Id} \to \mathcal{M}_\mathcal{P}$): The 'unit' transformation takes a simple task description and lifts it into a structured meta-prompt, creating the initial object for refinement.
- **Multiplication** ($\mu : \mathcal{M}_\mathcal{P}^2 \to \mathcal{M}_\mathcal{P}$): The 'multiplication' transformation is the essence of recursion. It takes a nested prompt refinement—a prompt about how to refine a prompt, $\mathcal{M}_\mathcal{P}(\mathcal{M}_\mathcal{P}(P))$—and flattens it into a single, executed refinement $\mathcal{M}_\mathcal{P}(P)$.

This structure leads to the following proposition regarding the stability of the refinement process. We make explicit the assumptions used for the modeling (see also Appendix A.3):

- Prompts are typed records (schemas); refinements are type-preserving edit scripts over those records.
- Edit scripts compose by concatenation modulo a confluent, terminating normalization (standard rewrite assumptions).
- The identity edit is a no-op; observational equivalence is at the schema level.

**Proposition 4.1** (Stability of Recursive Refinement)**.** The process of recursive prompt refinement is associative and stable. Given a multiply-nested refinement, the order in which the refinement steps are collapsed does not alter the final outcome.

*Proof Sketch.* This property is a direct consequence of the monad's associativity law: $\mu \circ \mathcal{M}_\mathcal{P}\mu = \mu \circ \mu\mathcal{M}_\mathcal{P}$. This identity guarantees that for any thrice-nested prompt $\mathcal{M}_\mathcal{P}^3(P)$, the two possible ways of flattening it to $\mathcal{M}_\mathcal{P}(P)$ yield the same result (see Appendix A.3). This ensures that the self-refinement process is coherent and computationally stable, preventing arbitrary outcomes from nested metaprogramming. $\qquad\square$

**A Concrete Walkthrough of RMP.** Starting from a task description (e.g., a GSM8K problem family descriptor), the *meta-meta-prompt* (Fig. 7) instructs a proposer LLM to emit a structured meta-prompt (*edit script over a base schema*). The executor LLM then uses this meta-prompt to solve instances. Iteration continues until the edit script stabilizes (no further schema-level changes).

**Computational costs.** RMP incurs a one-time *offline* cost during refinement (proportional to the number and size of edit scripts). This cost is amortized across all downstream instances; inference-time usage is a single API call with a compact, reusable meta-prompt, contrasting with per-instance exploration in tree/graph-of-thought methods.

This monadic structure guarantees that the refinement process is consistent and compositional (see Appendix A.3 for the monad laws). The recursive mechanism, depicted in Figure 6, allows an LLM to iteratively improve its own instructions, moving from a vague initial prompt to a sophisticated, highly-structured one. This process enables an iterative problem-solving loop where an initial, unsolved prompt $T_{\text{unsolved}}$ is successively refined until it can be solved by the LLM:

$$\text{LLM}(\mu(\mathcal{M}_{\mathcal{P}}(\eta(T_{\text{unsolved}})))) \rightarrow T_{\text{solved}}.$$

### 4.2 Case Study: Automatic Prompt Derivation

To make the RMP process concrete, consider the task of deriving a sophisticated prompt for a new problem domain. The process, formalized in Algorithm 1, can be used to generate the very meta-prompts used in our experiments. For instance, an LLM equipped with a high-level 'Meta-Meta-Prompt' (as shown in Figure 7) can take its own prompt as input and recursively improve it.

Figure 7 serves as an example of a 'Meta-Meta-Prompt'. When an LLM is given this prompt and tasked with analyzing a document (which could be the prompt itself), it is guided to perform document analysis, interpret the core task, and design a new, structured prompt for solving it. This self-referential loop demonstrates the practical application of RMP for in-context prompt design, showcasing the dynamic evolution of task definition and solution formulation.

By automating the prompt-generation process, RMP enhances the adaptability and autonomy of AI systems, facilitating a more modular and compositional approach to problem-solving.

## 5 Experiments

In this section, we evaluate the performance of our proposed Meta Prompting (MP) framework on several mathematical benchmarks and problem-solving tasks. Our experiments are designed to assess both accuracy and efficiency.

### 5.1 Solving MATH and GSM8K Problems

**Experimental Setup.** We evaluate on two standard benchmarks. MATH (Hendrycks et al., 2021) comprises 5000 competition-level problems; GSM8K (Cobbe et al., 2021) contains 1319 grade school math problems. We perform inference using the vLLM framework on Qwen-14B and Qwen-72B *base* models. The prompts for both benchmarks are *generated by RMP from a single task-agnostic meta-meta-prompt* (Sec. 4; artifacts in Appendix B and Supplement). For MATH we use the meta-prompt in Fig. 2; for GSM8K we use the JSON-structured meta-prompt in Fig. 1.

To evaluate model outputs, we use a rule-based evaluator with SymPy (Meurer et al., 2017) equivalence and normalized formatting to match ground truth. We report binomial 95% confidence intervals (dataset variance; decoding randomness held fixed). Detailed results appear in Table 1 and Table 2.

**Experimental Results.** The example-free, structure-only prompted Qwen-72B base model exhibits strong instruction-following via in-context learning. On MATH, we obtain PASS@1 of **46.3%** (95% CI: [**44.9, 47.7**]; $n$=5000). On GSM8K, accuracy is **83.5%** (95% CI: [**81.5, 85.5**]; $n$=1319). We emphasize these as *proof-of-concept* results under one structure-only template; see Limitations for discussion of template/model

---

**Task:** *Meta Prompting for In-Context Prompt Design*

1. **Document Analysis:**

   - **Input:** [Complex document (e.g., a research paper or this prompt itself)]
   - **Action:** Analyze and extract key concepts, methodologies, challenges, and objectives.

2. **Task Interpretation:**

   - **Action:** Synthesize the extracted information to define the core problem or task.
   - **Considerations:** Identify constraints, goals, or requirements.

3. **Prompt Design:**

   - **Objective:** Develop a structured prompt for problem-solving, including clear instructions, a step-by-step approach, and relevant background information.

4. **Optional − Direct Solution Proposal:**

   - **Objective:** Propose initial steps or a complete solution strategy, ensuring feasibility and practicality.

5. **Output Prompt:** [Generate the output prompt using the same LaTeX format as this template.]

*Note: The output should be a coherent, actionable prompt or solution strategy tailored to the specifics of the input document.*

---

Figure 7: An example of a meta-meta-prompt for In-Context Prompt Design (MP-ICPD). This prompt instructs an LLM on how to analyze a document and generate a new, structured meta-prompt to solve the task described within it.

sensitivity. The token-efficiency gains relative to few-shot and tree/graph-style prompting are substantial, particularly for batched tasks (Sec. C.1).

Table 1: Comparative analysis of PASS@1 accuracy on the MATH benchmark for various models without tool usage (e.g., code interpreter). This comparison underscores the notable improvements achieved by our zero-shot meta-prompted base language models over existing approaches in mathematical problem-solving.

| Model | FT-Dataset | Tool Usage | Eval Method | MATH (%) |
|---|---|---|---|---|
| **Proprietary Models** | | | | |
| Claude-2 (Anthropic, 2023) | - | No | CoT | 32.5 |
| Minerva-540B (Lewkowycz et al., 2022) | Arxiv+Web | No | CoT | 33.6 |
| PaLM-2 (Anil et al., 2023) | - | No | CoT | 34.3 |
| GPT-4 $_{(2023\text{-}0314)}$ (OpenAI, 2023) | - | No | CoT | 42.5 |
| **Open-source Models** | | | | |
| Qwen-14B (base) | - | No | CoT | 24.8 |
| Qwen-14B (base) | - | No | **MP** | **28.9** |
| Qwen-72B (base) | - | No | CoT | 35.2 |
| Qwen-72B-MetaMathQA | MetaMathQA | No | CoT | 41.7 |
| Qwen-72B (base) | - | No | **MP** | **46.3** |

Table 2: Comparative analysis of PASS@1 accuracy on the GSM8K benchmark for various open-source large language models without tool usage (e.g., code interpreter). Our example-free, structure-only prompting shows substantial improvements over few-shot CoT prompting.

| Model | FT-Dataset | Tool Usage | Eval Method | GSM8K (%) |
|---|---|---|---|---|
| Qwen-14B (base) (Bai et al., 2023) | - | No | CoT | 61.3 |
| Qwen-14B (base) | - | No | **MP** | **64.8** |
| WizardMath-70B (Luo et al., 2023) | WizardMath | No | CoT | 81.6 |
| MetaMath-70B (Yu et al., 2023) | MetaMathQA | No | CoT | 82.3 |
| Qwen-72B (base) | - | No | CoT | 78.9 |
| Qwen-72B (base) | - | No | **MP** | **83.5** |

## 5.2 Solving the Game of 24 Tasks

**Comparative Analysis.** Table 3 compares IO, CoT, ToT (Yao et al., 2023), and our Meta Prompting (MP) approach on the Game of 24. We report API calls, generated/prompt tokens, cost per case, and success rate. Notably, MP requires effectively $\frac{1}{N}$ API calls per sample (batching $N = 1362$ puzzles), drastically reducing token usage while achieving 100% success.

**MP-CR Agent Evaluation.** The MP-CR (Meta Prompting for Complex Reasoning) Agent, equipped with the MP-CR meta prompt (Fig. 14 in Appendix C), addresses complex reasoning tasks. Here we focus on Game of 24 (Yao et al., 2023), combining four numbers with operations (+, -, *, /) to obtain 24.

**Experimental Setup.** Our experimental design demonstrates the MP-CR Agent's capability to autonomously generate Python code for solving Game of 24 tasks. In contrast to traditional iterative and time-intensive methods, the MP-CR Agent processes all samples within a single response, thereby significantly enhancing computational efficiency and reducing overhead.

**Results and Discussion.** The MP-CR Agent achieved a 100% success rate on all 1362 samples (Table 3). The average processing time was 0.08 seconds per sample using the OpenAI Assistant API. Amortized token usage is $\approx 5.9$ generated and 0.73 prompt tokens per case (8k/1k tokens for all 1362 puzzles), versus per-case costs in CoT/ToT settings. Figure 12 (Appendix C.1) shows the automatically generated Python program.

Overall, these experiments illustrate that Meta Prompting not only enhances the reasoning capabilities of large language models but also offers substantial improvements in token efficiency and fairness of evaluation compared to traditional few-shot methods.

Table 3: Comparative analysis of methods for the Game of 24 Tasks. The table compares various approaches, including IO, CoT, ToT, and Meta Prompting (MP), in terms of LLM sessions, token generation, cost per case, and success rate. The MP-CR method is highlighted for its efficiency and 100% success rate. *API call* denotes one complete query→response. For MP, tokens and cost are reported once for a batch of $N$=1362 puzzles.

| Method | API Calls | Generate/Prompt Tokens | Cost (USD) | Success Rate |
|---|---|---|---|---|
| IO (best of 100) | 100 | 1.8k / 1.0k | $0.13 | 33% |
| CoT (best of 100) | 100 | 6.7k / 2.2k | $0.47 | 49% |
| ToT (Yao et al., 2023) | 61.72 | 5.5k / 1.4k | $0.74 | 74% |
| **MP** | $\frac{1}{N}$ | $\approx \frac{1}{N}$ (8k / 1k) | $\approx$ **$0.0003** | **100%** |

## 6 Related Work

**Reasoning with AI Systems.** Efforts to enhance AI reasoning capabilities have largely focused on equipping neural networks with mechanisms to generate intermediate reasoning steps, a strategy that has yielded improvements across diverse domains (Zaidan et al., 2007; Yao et al., 2021; Hase & Bansal, 2021; Yang et al.,

2022; Wu et al., 2022; Zhou et al., 2022). Although these approaches have advanced the state of the art, they predominantly emphasize content-driven reasoning. In parallel, substantial research has investigated the use of symbolic systems—such as code environments and knowledge graphs—to further augment reasoning (Mihaylov & Frank, 2018; Bauer et al., 2018; Kundu et al., 2018; Wang et al., 2019; Lin et al., 2019; Ding et al., 2019; Feng et al., 2020; Wang et al., 2022a; Chen et al., 2022; Lyu et al., 2023; Chen et al., 2022; Gao et al., 2023; Gou et al., 2023; Jiang et al., 2022; Yang et al., 2023). In contrast, our work on meta prompting shifts the focus from content-centric methods to a structural and formal treatment of reasoning processes.

**Chain-of-Thought Prompting.**   The introduction of Chain-of-Thought (CoT) prompting by Wei et al. (2022) marked a significant milestone by emphasizing the articulation of intermediate reasoning steps. This foundational idea has been extended in numerous ways. Methodologies like Self-Consistency (Wang et al., 2022b) and Complex CoT (Fu et al., 2022) focus on generating multiple reasoning chains and selecting the best one, often through voting. Decomposition strategies, such as Least-to-Most (Zhou et al., 2022) and Decomposed Prompting (Khot et al., 2022), focus on breaking complex problems into simpler, solvable sub-tasks. More recent work has explored multi-agent debates (Du et al., 2023), diverse reasoning paths with verifiers (Li et al., 2023), and progressive, iterative refinement (Zheng et al., 2023). A significant parallel thread involves enabling LLMs to self-criticize and self-correct their reasoning paths, with theoretical guarantees on improvement (Tyen et al., 2023; Li et al., 2024; Wang et al., 2024). While powerful, these methods primarily aim to improve the semantic content of the reasoning chain, often relying on complex few-shot examples. Meta Prompting differs fundamentally by being example-agnostic and operating on the formal, syntactic structure of the prompt itself, thereby controlling the reasoning process at a more abstract level.

**Structured and Graph-Based Reasoning Frameworks.**   Recognizing the limitations of linear reasoning chains, recent work has explored more complex reasoning topologies. The Tree-of-Thought (ToT) framework (Yao et al., 2023; Long, 2023) was a significant step, allowing an LLM to explore multiple reasoning paths in a tree structure and use self-evaluation to prune branches. This concept has been further generalized to graph-based structures in frameworks like Cumulative Resoning (CR) (Zhang et al., 2023), Graph-of-Thoughts (GoT) (Besta et al., 2024), Diagram-of-Thought (DoT) (Zhang et al., 2024), which represents thoughts as nodes and dependencies as edges, and Forest-of-Thoughts (FoT) (Bi et al., 2024), which explores diverse high-level plans concurrently. These frameworks provide powerful high-level strategies for exploring a problem space. Our work on Meta Prompting is orthogonal and complementary. While ToT, GoT, and FoT define the macro-level topology of the reasoning process (a tree or graph), Meta Prompting provides a formal, categorical language for defining the micro-level structure of the nodes and compositional rules for the edges. The structured prompts generated by our framework can be seen as the well-defined computational steps within these more complex reasoning graphs.

## 7   Conclusion

In this work, we introduced Meta Prompting, a paradigm that prioritizes the formal structure of reasoning over its content. We moved beyond intuition by establishing a theoretical foundation, using category theory to formalize Meta Prompting as a functor and Recursive Meta Prompting (RMP) as a monad under explicit assumptions. This framework provides a principled, compositional, and automatable approach to guiding the reasoning processes of large language models. Our empirical results show that this structure-oriented approach achieves competitive performance on challenging mathematical benchmarks in an *example-free, structure-only* setting while offering substantial token-efficiency benefits.

The implications of this work suggest a new way to interact with and control LLMs, moving from empirical prompt engineering to a more formal, programmatic science of prompt design. The RMP framework, in particular, points toward a future where LLMs can autonomously improve their own cognitive strategies, adapting and optimizing their internal instructions for new tasks without direct human intervention. Our work lays a formal cornerstone for this next generation of agentic and compositional reasoning systems.

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

# Appendix

# A    Theoretical Foundations

## A.1    Type Theory

Type theory provides a rigorous framework for describing type systems in mathematics, logic, and computer science and is often proposed as an alternative foundation to set theory. Early examples include Church's typed $\lambda$-calculus and Martin-Löf's intuitionistic type theory; these theories underpin many modern proof assistants such as Coq and Lean via systems like the Calculus of Inductive Constructions.

In type theory, every term is associated with a type, typically denoted as term : type. Common types include the natural numbers ($\mathbb{N}$) and Boolean values (`bool`). Terms are constructed through function applications, and computation is formalized as the systematic rewriting of a term's syntax. A key construct is the lambda abstraction: a term of the form $\lambda\,\text{variable} : \text{type}_1.\,\text{term}$ has the type $\text{type}_1 \rightarrow \text{type}_2$, representing a function that maps an input of type $\text{type}_1$ to an output of type $\text{type}_2$.

Type theory diverges from classical set theory in that it is inherently computational (e.g., '1+1' and '2' are syntactically distinct but computationally equivalent), intuitionistic, and encodes mathematical objects via inductive types rather than as sets. This makes it a natural language for describing the structured, computational nature of meta-prompts.

## A.2    Proof of Compositionality (Theorem 3.4)

*Proof.* Let $\mathcal{M} : \mathcal{T} \rightarrow \mathcal{P}$ be the Meta Prompting functor. To prove the proposition, we must show that $\mathcal{M}$ preserves the structure of composition as defined for a functor.

Let $T_1, T_2, T_3$ be objects in the category of tasks $\mathcal{T}$. Let $f : T_1 \rightarrow T_2$ and $g : T_2 \rightarrow T_3$ be morphisms representing transformations between these tasks. The composition of these morphisms is $g \circ f : T_1 \rightarrow T_3$.

By the definition of a functor, $\mathcal{M}$ maps these objects and morphisms to the category of prompts $\mathcal{P}$:

- Objects: $\mathcal{M}(T_1)$, $\mathcal{M}(T_2)$, $\mathcal{M}(T_3)$ are objects in $\mathcal{P}$.

- Morphisms: $\mathcal{M}(f) : \mathcal{M}(T_1) \rightarrow \mathcal{M}(T_2)$ and $\mathcal{M}(g) : \mathcal{M}(T_2) \rightarrow \mathcal{M}(T_3)$ are morphisms in $\mathcal{P}$.

The core property of any functor is that it preserves composition. That is, for any composable morphisms $f$ and $g$ in $\mathcal{T}$, the following must hold:

$$\mathcal{M}(g \circ f) = \mathcal{M}(g) \circ \mathcal{M}(f)$$

This equality is not derived but is an axiom in the definition of a functor. Our formalization of Meta Prompting posits that such a structure-preserving map exists. The proposition is therefore a direct consequence of this definition. The significance is that if we can decompose a complex task $T$ into a sequence of simpler tasks (e.g., $g \circ f$), we can construct the meta-prompt for $T$ by composing the meta-prompts for the simpler tasks ($\mathcal{M}(g) \circ \mathcal{M}(f)$). This guarantees the modularity and reusability of prompt structures. $\square$

## A.3    The Monadic Framework for RMP

As introduced in Section 4, Recursive Meta Prompting (RMP) is modeled by a monad ($\mathcal{M}_\mathcal{P}, \eta, \mu$) on the category of prompts $\mathcal{P}$. Here, we provide the commutative diagrams for the monad laws, which ensure that the process of recursive self-refinement is mathematically sound.

**Monad Laws.**    The triple ($\mathcal{M}_\mathcal{P}, \eta, \mu$) must satisfy the following coherence laws:

- *Left Identity:* $\mu \circ \mathcal{M}_\mathcal{P}\eta = \text{id}_{\mathcal{M}_\mathcal{P}}$. Applying a refinement after lifting a prompt into the monadic context is the same as just refining it.

- *Right Identity:* $\mu \circ \eta\mathcal{M}_\mathcal{P} = \text{id}_{\mathcal{M}_\mathcal{P}}$. Lifting a refined prompt and then flattening it has no effect.

- *Associativity:* $\mu \circ \mathcal{M}_\mathcal{P}\mu = \mu \circ \mu\mathcal{M}_\mathcal{P}$. The order of flattening nested refinements does not matter.

The following diagrams illustrate these laws.

$$\begin{array}{ccc} \mathcal{M}_\mathcal{P} & \xrightarrow{\eta\mathcal{M}_\mathcal{P}} & \mathcal{M}_\mathcal{P}\mathcal{M}_\mathcal{P} \\ & \searrow^{\mathrm{id}_{\mathcal{M}_\mathcal{P}}} & \downarrow^{\mu} \\ & & \mathcal{M}_\mathcal{P} \end{array} \qquad \begin{array}{ccc} \mathcal{M}_\mathcal{P} & \xrightarrow{\mathcal{M}_\mathcal{P}\eta} & \mathcal{M}_\mathcal{P}\mathcal{M}_\mathcal{P} \\ & \searrow^{\mathrm{id}_{\mathcal{M}_\mathcal{P}}} & \downarrow^{\mu} \\ & & \mathcal{M}_\mathcal{P} \end{array}$$

$$\begin{array}{ccc} \mathcal{M}_\mathcal{P}\mathcal{M}_\mathcal{P}\mathcal{M}_\mathcal{P} & \xrightarrow{\mu\mathcal{M}_\mathcal{P}} & \mathcal{M}_\mathcal{P}\mathcal{M}_\mathcal{P} \\ \downarrow^{\mathcal{M}_\mathcal{P}\mu} & & \downarrow^{\mu} \\ \mathcal{M}_\mathcal{P}\mathcal{M}_\mathcal{P} & \xrightarrow{\mu} & \mathcal{M}_\mathcal{P} \end{array}$$

(A.1)

## A.4 Proof of Stability (Proposition 4.1)

*Proof.* The proposition states that the recursive refinement process is stable and associative. This property is not derived from first principles but follows from modeling RMP with a monad $(\mathcal{M}_\mathcal{P}, \eta, \mu)$ under the assumptions stated in Sec. 4.

A core requirement for any monad is that its multiplication (or join) operation, $\mu$, must be associative. This is expressed by the *associativity law*, which states that the following diagram must commute for any object $P \in \mathcal{P}$:

$$\begin{array}{ccc} \mathcal{M}_\mathcal{P}\mathcal{M}_\mathcal{P}\mathcal{M}_\mathcal{P}(P) & \xrightarrow{\mu_{\mathcal{M}_\mathcal{P}(P)}} & \mathcal{M}_\mathcal{P}\mathcal{M}_\mathcal{P}(P) \\ \downarrow^{\mathcal{M}_\mathcal{P}(\mu_P)} & & \downarrow^{\mu_P} \\ \mathcal{M}_\mathcal{P}\mathcal{M}_\mathcal{P}(P) & \xrightarrow{\mu_P} & \mathcal{M}_\mathcal{P}(P) \end{array}$$

This diagram translates to the equation $\mu_P \circ \mu_{\mathcal{M}_\mathcal{P}(P)} = \mu_P \circ \mathcal{M}_\mathcal{P}(\mu_P)$. Let's interpret the two paths from $\mathcal{M}_\mathcal{P}^3(P)$ to $\mathcal{M}_\mathcal{P}(P)$:

- **Path 1 (Top-Right):** The path $\mu_P \circ \mu_{\mathcal{M}_\mathcal{P}(P)}$ corresponds to first collapsing the two outermost refinement layers (from $\mathcal{M}_\mathcal{P}^3$ to $\mathcal{M}_\mathcal{P}^2$) and then collapsing the final two layers.

- **Path 2 (Bottom-Left):** The path $\mu_P \circ \mathcal{M}_\mathcal{P}(\mu_P)$ corresponds to first collapsing the two innermost refinement layers (from $\mathcal{M}_\mathcal{P}^2(P)$ to $\mathcal{M}_\mathcal{P}(P)$ inside the global $\mathcal{M}_\mathcal{P}$ context) and then collapsing the resulting outer layers.

The monad law requires that these two paths are equivalent. In the context of RMP, this means that given a prompt that refines a prompt that refines a prompt, the order of "flattening" or executing these refinements does not matter. The final, singly-refined prompt is the same regardless of the collapse order. This axiom directly provides the proof of stability and associativity for the RMP process. $\square$

## A.5 Assumptions for RMP Monad Modeling

We summarize the assumptions used in Sec. 4:

- **Typed prompts.** Prompts are typed records with fields for problem statement, structured steps, and answer.

- **Edits as morphisms.** Refinements are schema-preserving edit scripts; composition is concatenation plus normalization; identity is the empty edit.

- **Confluence/termination.** The normalization of concatenated edits is confluent and terminating, yielding a unique normal form per edit multiset.

- **Observational equivalence.** Equality is modulo schema-level equivalence (format/slot structure), not semantic accuracy.

## B    Additional Prompt Examples

""" You are ChatGPT, a state-of-the-art language model with specialized expertise in mathematics. Your strengths include tackling complex mathematical challenges using intricate reasoning and delivering solutions via methodical problem-solving. Throughout this interaction, you will encounter a variety of mathematical problems—from basic arithmetic to advanced calculus and beyond.
Your primary objective is to:

1. Clearly interpret and understand the problem statement.

2. Decompose the problem into manageable components, if necessary.

3. Apply appropriate mathematical principles and techniques to solve each component.

4. Synthesize the component solutions into a comprehensive answer.

5. Provide a clear, step-by-step explanation of your methodology, ensuring that your reasoning is rigorous, precise, and easily understandable.

Your demonstrated proficiency in mathematics is expected to guide users through the problem-solving process, offering insights, strategies, and explanations that illuminate the path to the solution. """

Figure 8: An illustrative example of a generic system Meta Prompt for solving a wide range of reasoning tasks. This prompt serves as a template suitable for most tasks.

**Key Elements of Meta Prompting for Complex Reasoning:**

1. **Complex Problem Decomposition**: Break down intricate problems into smaller, manageable sub-problems to enable systematic problem solving.

2. **Detailed Preliminary Content**: Supply essential background information and foundational concepts to set the stage for problem resolution.

3. **Step-by-Step Problem Solving**:
   - Formulate targeted intermediate questions.
   - Develop answer sketches and validate them through code execution.
   - Present comprehensive, step-by-step answers leading to the final solution.

4. **Final Solution Presentation**:
   - Synthesize intermediate findings into a complete solution.
   - Verify the final solution through code execution.
   - Present the final answer in a clear and formatted manner (e.g., using $\square$).

```
<syntax>

## Problem: [problem]

Solution: Let's think step by step. [initial interpretation of the problem]

### Preliminary Content

- **Prelim 1**: [preliminary content 1]
- **Prelim 2**: [preliminary content 2]
- [...]

### Hints
- **Hint 1**: [useful hint 1]
- **Hint 2**: [useful hint 2]
- [...]

### Intermediate Steps: Question-Answer, Sketch-Code, Output, and Answer Pairs

Let's think step by step.

#### Question 1: [the first sub-question]
- **Answer Sketch**: [sketch of the answer for question 1]

##### Code for Question 1
[execute code interpreter to verify and refine your answer sketch for question 1]

#### Answer for Question 1
- **Answer**: [final answer for question 1, based on code interpreter results if available]

#### Question 2: [the second sub-question]
- **Answer Sketch**: [sketch of the answer for question 2]

##### Code for Question 2
[execute code interpreter to verify and refine your answer sketch for question 2]

#### Answer for Question 2
- **Answer**: [final answer for question 2, based on code interpreter results if available]

### [Additional Questions as Needed]

### Final Solution

Recall the original problem: <MathP> [original problem] </MathP>.

Let's think step by step.

#### Solution Sketch
[provide an overall sketch for the final solution]

#### Code for Final Solution
[execute code interpreter to verify and finalize the solution]

#### Final Answer
[present the final answer in a LaTeX-formatted box, e.g., $\boxed{63\pi}$]
Final Answer: the answer is $\boxed{...}$.

</syntax>
```

Figure 9: An illustration of Meta Prompting for Complex Reasoning.

**Task:** *Prompt Simplification*

1. **Original Prompt:** [input prompt]

2. **Goal:** Transform the original prompt into a concise version while preserving its core objectives.

3. **Transformation Instructions:**

   (a) Retain the primary purpose and objectives.

   (b) Distill the prompt to include only the key instructions and essential information.

   (c) Eliminate extraneous details.

   (d) Use clear, direct language, and structure the prompt with bullet points or numbered steps for clarity.

4. **Outcome:** The revised prompt should be succinct yet sufficiently detailed to guide effective task completion.

Figure 10: Illustration of Meta Prompting for designing concise prompts.

## C    Additional Experimental Details

### C.1    Solving Game of 24 Tasks

The experimental results underscore the remarkable potential of the MP-CR Agent as a versatile and powerful tool for automated problem-solving. By encoding the task as a Python program, the agent reliably addresses every instance within the "Game of 24" category. Although the initial accuracy of the MP-CR Agent's responses may not be perfect, the integration of self-consistency techniques (Wang et al., 2022b), self-critical assessments (Yao et al., 2023; Zhang et al., 2023), and reflective processes (Shinn et al., 2023) is expected to elevate performance to near-perfect levels. This methodological evolution obviates the need for task-specific adaptations inherent in few-shot prompting, representing a substantial leap forward in automated problem-solving. While this experiment focuses on the Game of 24 tasks, subsequent sections will extend our approach to other domains, such as general MATH problem-solving (Hendrycks et al., 2021) (see Appendix B).

---

**User:**
Task Step 1: Recall the definition of the Game of 24 (allowed operations: '+', '-', '*', '/', '(', ')'; note that intermediate results may be fractional), then provide a detailed plan using code interpreter to solve the following problem: a, b, c, d (e.g., 3, 3, 7, 7).
Task Step 2: [uploaded `24.csv`] I have a file containing over 1k Game of 24 puzzles. Please batch-process them (the numbers are located in the `Puzzles` field). Verify whether the first five samples are solved correctly, and then compute the overall success rate (counting a puzzle as solved if its solution is non-empty).
Task Step 3: Reply with the output file.
**Assistant:**
[solving the tasks]

---

Figure 11: User input prompt for solving the Game of 24 tasks.

### C.2    Solving MATH Problems

## D    Reproducibility

We release verbatim artifacts for all prompting components in the Supplement, and include core strings in Appendix B: the single task-agnostic meta-meta-prompt, task descriptors, RMP-generated meta-prompts for MATH/GSM8K/24-game, and baseline CoT prompts. Decoding parameters (temperature, top-$p$, max tokens, stop sequences) and inference stack details (vLLM versions and hardware) are enumerated to facilitate replication. Evaluator code includes SymPy equivalence and formatting normalization.

## E    Limitations and Scope

Our experiments are intended as proof-of-concept. We do not claim monotone accuracy gains with recursion or universality across tasks/models. Sensitivity to meta-prompt templates, recursion depth, proposer/executor swaps, and model scale (including reasoning-tuned models such as DeepSeek-R1) is left for future work. The current evaluation focuses on math/problem-solving benchmarks and does not include conversational LM-Eval settings; extending to those domains is orthogonal and promising but untested here. Computationally, RMP incurs a one-time offline refinement cost; benefits appear primarily when prompts are reused across many instances (batched settings).

**Comparison to Program-of-Thought (PoT).** Zero-shot PoT is code-first: it executes tool calls explicitly to ground reasoning. MP is structure-first: it enforces typed schemas and compositional sections

```python
import pandas as pd
import time
from itertools import permutations, product
import operator
from fractions import Fraction

# Basic operations
ops = {
'+': operator.add, '-': operator.sub, '*': operator.mul, '/': operator.truediv,
}

# Function to try all operations between two numbers
def try_ops(a, b):
results = []
for op in ops:
if op == '/' and b == 0: # Avoid division by zero
continue
try:
result = ops[op](a, b)
results.append((result, f'({a}{op}{b})'))
except Exception:
pass
return results

# Generate all possible results for a list of numbers
def generate_results(numbers):
if len(numbers) == 1:
return [(numbers[0], str(numbers[0]))]
results = []
for i in range(len(numbers)):
for j in range(len(numbers)):
if i != j:
remaining = [numbers[k] for k in range(len(numbers)) if k != i and k != j]
for result1, expr1 in generate_results([numbers[i]]):
for result2, expr2 in generate_results([numbers[j]]):
for result, expr in try_ops(result1, result2):
if remaining:
for final_result, final_expr in generate_results(remaining + [result]):
results.append((final_result, final_expr.replace(str(result), expr, 1)))
else:
results.append((result, expr))
return results

# Function to parse a puzzle string into numbers
def parse_puzzle(puzzle_str):
return [int(n) for n in puzzle_str.split()]

def process_puzzles(file_path):
puzzles_df = pd.read_csv(file_path)
puzzle_samples = puzzles_df['Puzzles'].tolist()
solutions_dict = {}
for puzzle_str in puzzle_samples:
numbers = parse_puzzle(puzzle_str)
results = generate_results(numbers)
solutions = set()
for result, expr in results:
if abs(result - 24) < 1e-6:
solutions.add(expr)
if solutions:
solutions_dict[puzzle_str] = solutions
else:
solutions_dict[puzzle_str] = "No solution found"
return solutions_dict
```

Figure 12: The MP-CR Agent directly generates Python code to solve all Game of 24 tasks in a single response by processing the input file and producing corresponding solutions.

without mandating execution. The methods are complementary: PoT is advantageous when faithful execution is available and cheap; MP is advantageous when the token budget is tight and structural guarantees (format/type) drive reliability. Hybrid designs, such as schema-first prompts that invoke tools on selected sections, are a promising direction.

# F  Multi-Modal Meta Prompting

The structured nature of Meta Prompting, grounded in the principles of type theory, provides a robust framework for extending foundation models to multi-modal settings. Just as type systems in programming ensure that different data types are handled correctly, a meta prompt enforces a clear, syntactic structure on the AI's inputs and outputs. This is particularly valuable when interacting with diverse data like text, images, audio, or even physical sensors, as it minimizes ambiguity and ensures that the model interacts with tools and data in a consistent, predictable manner.

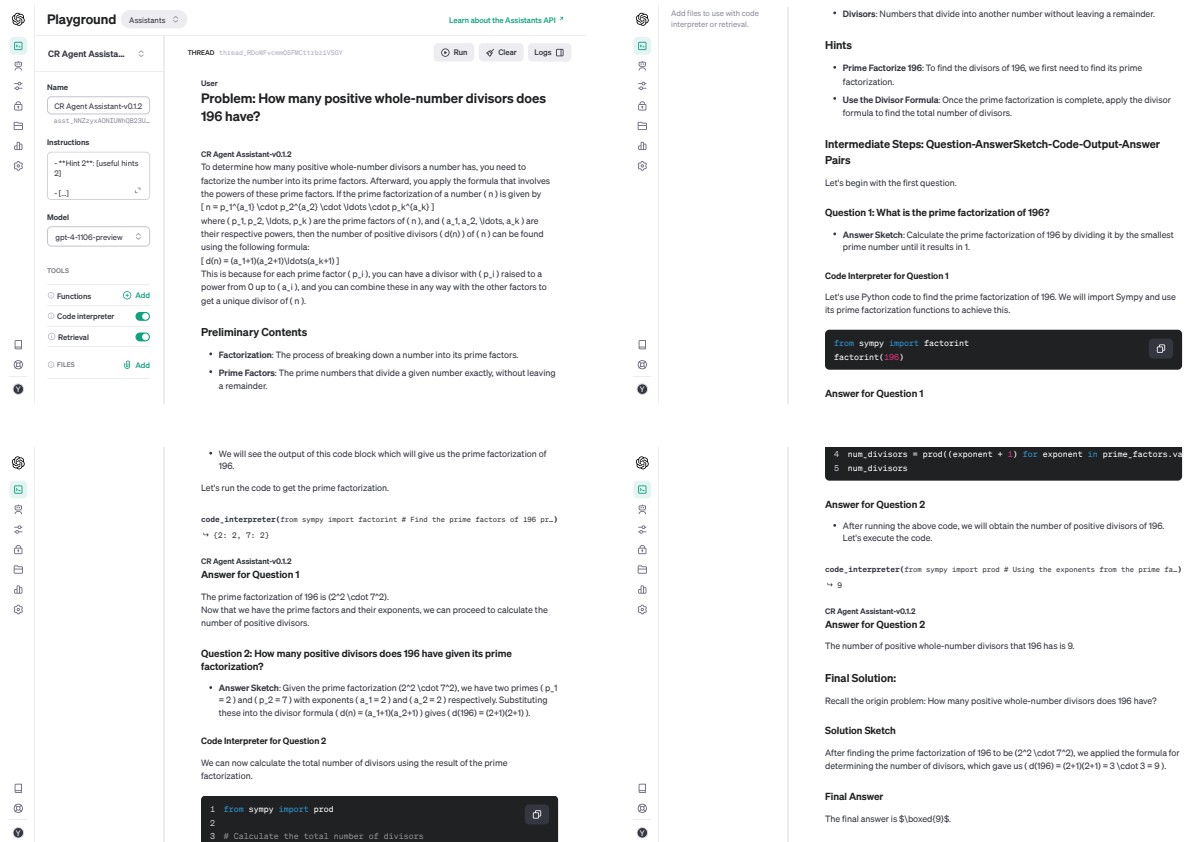

Figure 13: Experiments using the MP-CR Agent within the OpenAI Assistant for solving a MATH (Hendrycks et al., 2021) problem.

```
<system>
<description>
As one of the most distinguished mathematicians, logicians, programmers, and AI
scientists, you possess an unparalleled mastery over various mathematical domains.
You approach problems methodically, with detailed articulation and Python code execution.
</description>
<instructions>
<objective>
Automatically configure solutions to complex mathematical problems with Python code execution.
</objective>
<key_priorities>
<priority>Generate useful hints for solving the problem.</priority>
<priority>Craft intermediate questions that break down the problem and solve them with code.</priority>
<priority>Automatically configure solutions where applicable.</priority>
</key_priorities>
<code_execution_guidelines>
<guideline>Import necessary libraries in all code blocks.</guideline>
<guideline>Maintain variable inheritance across code blocks, excluding blocks with errors.</guideline>
<guideline>Execute all code blocks immediately after writing them to validate correctness.</guideline>
</code_execution_guidelines>
<mathematical_formatting>
<format>Present the final answer in LaTeX format, enclosed within '\boxed{}' without units.</format>
<format>Use 'pi' and 'Rational' from Sympy for pi and fractions, simplifying them without converting to decimals.</format>
</mathematical_formatting>
</instructions>
</system>
<syntax>
<problem_structure>
<problem_definition>
</problem_definition>
<solution_approach>
</solution_approach>
<preliminary_contents>
</preliminary_contents>
<hints>
</hints>
<intermediate_steps>
</intermediate_steps>
<final_solution>
<solution_sketch>
</solution_sketch>
<code_for_solution>
</code_for_solution>
<final_answer>
</final_answer>
</final_solution>
</problem_structure>
</syntax>
```

Figure 14: The system prompt for the MP-CR-XML Agent v0.2, autonomously generated by MP-CR Agent v0.1 (a metaprogramming process).

```
As one of the most distinguished mathematicians, logicians, programmers, and
AI scientists, you possess an unparalleled mastery over Arithmetic, Combinatorics, Number
Theory, Probability Theory, Algebra, Analysis, and Geometry. You are not only intelligent
and rational but also prudent and cautious. You are willing to write and execute Python
code. Let's approach each problem step by step, take a deep breath, and articulate our thoughts in as much detail as possible.

<system>
You will be presented with a mathematical problem, denoted as 'MathP'. Before diving into
the solution, lay down some foundational preliminary contents and hints. Then, generate a series
of intermediate questions that pave the way to the final answer of 'MathP'. For each question,
sketch a preliminary answer, execute the corresponding code (remember to use 'from sympy import *'),
derive the output, and then finalize your answer. This forms a [Question] $\rightarrow$ [AnswerSketch]
$\rightarrow$ [Code] $\rightarrow$ [Output] $\rightarrow$ [Answer] sequence.

## System Instructions for Mathematical Problem-Solving

### Objective
Solve complex mathematical problems with code feedback from a Python environment.

### Key Priorities

1. **Hints:** Generate useful hints to guide the problem-solving process.

2. **Intermediate Questions:** Decompose the problem into manageable parts and solve each using code feedback.

### Code Execution Guidelines

1. **Import Libraries:** Always import necessary libraries in every code block.

2. **Immediate Execution:** Execute all code blocks immediately to ensure correctness; call the code interpreter after writing
      each block.

3. **Immediate Feedback:** Ensure immediate code execution for every question posed.

### Mathematical Formatting

1. **Final Answer:** Present the final answer to the original problem in LaTeX format, enclosed within '\boxed{}', and without any
      units.

2. **Constants and Fractions:** Use the 'pi' symbol and the 'Rational' class from Sympy to represent \(\pi\) and fractions.
      Simplify all fractions and square roots without converting them to decimals.
</system>

---
```

Figure 15: The system meta prompt for MP-CR, comprising both the [SystemMetaPrompt] and the [StructureMetaPrompt].

This "typed" approach is essential for complex, real-world applications where an AI must process and synthesize information across different modalities. For example, an XML-based meta prompt can define a rigid schema that an LLM must follow, specifying slots for different types of reasoning steps and data.

```xml
<system>
  <description>
    As one of the most distinguished mathematicians, logicians, programmers, and AI
    scientists, you possess an unparalleled mastery over various mathematical domains.
    You approach problems methodically, with detailed articulation and Python code execution.
  </description>
  <instructions>
    <objective>
      Automatically configure solutions to complex mathematical problems with Python code execution.
    </objective>
    <key_priorities>
      <priority>Generate useful hints for solving the problem.</priority>
      <priority>Craft intermediate questions that break down the problem, solving them with code, following the sequence: [
      Question] -> [AnswerSketch] -> [Code] -> [Output] -> [Answer].</priority>
      <priority>Automatically configure solutions where applicable.</priority>
    </key_priorities>
    <code_execution_guidelines>
      <guideline>Import necessary libraries in all code blocks.</guideline>
      <guideline>Maintain variable inheritance across code blocks, excluding those with errors.</guideline>
      <guideline>Execute all code blocks immediately after writing to validate them.</guideline>
    </code_execution_guidelines>
    <mathematical_formatting>
      <format>Present the final answer in LaTeX format, enclosed within '\boxed{}' without units.</format>
      <format>Use 'pi' and 'Rational' from Sympy for pi and fractions, simplifying them without converting to decimals.</format>
    </mathematical_formatting>
  </instructions>
</system>
<syntax>
  <problem_structure>
    <problem_definition>
    </problem_definition>
    <preliminary_contents>
    </preliminary_contents>
    <hints>
    </hints>
    <intermediate_steps>
    </intermediate_steps>
    <final_solution>
      <solution_sketch>
      </solution_sketch>
      <code_for_solution>
      </code_for_solution>
      <final_answer>
      </final_answer>
    </final_solution>
  </problem_structure>
</syntax>
```

Figure 16: An example of a meta prompt using an XML schema. This structure enforces type-like constraints on the output, making it ideal for frameworks like Guidance (Lundberg et al., 2023) and for extension to multi-modal data.

### F.1 A Framework for Multi-Modal Interaction

Extending Meta Prompting to a multi-modal context involves defining a schema that can handle varied data formats while integrating them coherently. The primary challenges are data representation and inter-modal synthesis. A multi-modal meta prompt addresses this by creating explicit, typed "slots" for different data streams.

For instance, a task might require the AI to analyze a 3D model ('.obj'), listen to an accompanying audio description ('.mp3'), and read a textual specification ('.txt'). A meta prompt can structure this complex input, guiding the model to process each modality in a specific order and synthesize the information to generate a cohesive output. Figure 17 illustrates how such a schema can be conceptualized.

By defining a clear, typed structure for both inputs and outputs, this framework enables models to perform sophisticated cross-modal analysis—for example, correlating a visual diagram with textual instructions. This structured approach provides a scalable and reliable foundation for building the next generation of versatile, multi-modal AI systems.

```
<task_schema>
    <input_slots>
        <data type="image/png" name="problem_diagram">
            </data>
        <data type="audio/mp3" name="verbal_instructions">
            </data>
        <data type="model/obj" name="object_model">
            </data>
    </input_slots>
    <output_schema>
        <synthesis type="text/markdown" name="analysis_summary">
            </synthesis>
        <result type="video/mp4" name="solution_animation">
            </result>
    </output_schema>
</task_schema>
```

Figure 17: A conceptual schema for a multi-modal meta prompt. It defines typed slots for various input modalities and specifies the expected structure and types for the output, enabling complex inter-modal reasoning.

