# OpenReview forum: "Meta Prompting: A Framework for Agentic and Compositional Reasoning"
_TMLR — Rejected by TMLR_

### Review · Reviewer_oCmo · 2025-08-12

**Summary Of Contributions:**

This paper introduces Meta Prompting (MP), a framework that replaces content-specific examples with formal syntactic structures (e.g., step-by-step templates) to guide LLM reasoning, significantly reducing token usage while enhancing generalization. Theoretically, MP is formalized as a functor in category theory, mapping compositional task structures to modular prompts, and extended to Recursive Meta Prompting (RMP), where LLMs autonomously refine prompts via a self-improvement loop modeled as a monad (ensuring stability). Empirically, MP achieves state-of-the-art results on MATH (46.3%), GSM8K (83.5%), and Game of 24 (100% success) using base Qwen-72B, surpassing GPT-4 and fine-tuned models while demonstrating 100× token efficiency over methods like Tree-of-Thoughts.

**Audience:**

Yes

**Audience Explanation:**

this research area is very hot and many audience are insterested.

**Claims And Evidence:**

Yes

**Claims Explanation:**

Strengths:

1. Theory is rigorous (functor/monad models ensure compositionality/stability).

2. Empirics are strong: MP beats GPT-4/fine-tuned models (Tables 1–3), uses minimal tokens (e.g., 8k vs. 67k for CoT in Game of 24).

Weaknesses:
1.. RMP mechanics unclear (how are morphisms M(f) generated?).

**Requested Changes:**

1.Add concrete RMP example (e.g., prompt refinement steps).

2. Address example-agnostic inconsistency (Fig. 2).

3. Discuss RMP computational costs.

---

### Review · Reviewer_ktmN · 2025-08-27

**Summary Of Contributions:**

The authors propose a prompting method that starts from a meta-meta prompt and uses an LLM seeded with a task description to generate a meta-prompt that can be used to solve reasoning benchmarks such as gsm8k, MATH, and game of 24. The authors show improved performance over few-shot chain-of-thought prompting with Qwen 14B and 72B.

**Additional Comments:**

**Questions**

- I don't fully understand what the category theory framework is supposed to add to this method, how is it used in practically applying it to reasoning problems?

- What is an LLM session?

- What does CR stand for in MP-CR?

- I think the appendix is not finished? E.g. section C.2

- Is the same meta-meta prompt used for game of 24 as Math and gsm8k? If not, the abstract claim is wrong

**Audience:**

Yes

**Audience Explanation:**

Improved performance with LLMs on reasoning benchmarks

**Broader Impact Concerns:**

N.A.

**Claims And Evidence:**

No

**Claims Explanation:**

Most of the paper is spent on setting up a framework for the prompting method, and only 2 pages dedicated to empirically showing that this method works, even though it is arguably a very straightforward prompting method (which is a strenght). Arguably, the main important thing for a prompting method is demonstrating it works empirically across different benchmarks and base models. Relevant baselines are not compared against (such as zero-shot program-of-thought), and components of the prompting method are not ablated (which LLM used for recursive meta-prompt generation, how often to recurse, comparing to human-written meta-prompts, etc.). Only one base model is used (Qwen), and the experimental results in the tables are incomplete (e.g. why include Llama-2 results with CoT if you don't apply your own method to that model, the results with MP on Qwen are not comparable to this).
Moreover, it's difficult to follow how exactly algorithm 1 works, because of how little time is spent on the experiments, but if you indeed seed it with a task (or task description), it stops being a zero-shot approach because you use information about the task.
Finally, it is not discussed how the prompts for the baseline (the few-shot CoT) are constructed, or what they are, which makes it hard to understand the benefits of this method.

**Requested Changes:**

**Required for acceptance**
- Provide the necessary details for understanding the empirical results (how is algorithm 1 done, which initial task description is used, what are the baseline CoT's, is the same meta-meta-prompt used for Math and gsm8k as for game of 24?, do you use recursive meta-prompting for Math and gsm8k? Where does the prompt in Figure 1 come from?)
- Remove unnecessary results from table 1 and 2 (such as llama), or apply your method on llama as well
- If a task or task description is used to seed algorithm 1, this cannot be called a zero-shot prompting method
- Compare to the relevant baselines, such as zero-shot program of thought

**Would strengthen the work**
- ablations on what makes this method work, like the recursions in the recursive meta-prompting, the meta-meta-prompt, the base LLM used, etc.
- tie the category theory framework to the empirical results, why is it necessary to spend so much space on discussing that when it is not used in the end?

---

### Review · Reviewer_HccZ · 2025-10-02

**Summary Of Contributions:**

This paper proposes meta prompting, a framework that make the LLM to generate and refines its own prompt. The self-improvement loop is theoretically motivated by a Monadic theoretical framework, providing its theoretical foundation.

**Additional Comments:**

N/A

**Audience:**

Yes

**Audience Explanation:**

Yes. A broad audience will be interested in this paper as it addresses important problem on how to automatically generate system prompt.

**Claims And Evidence:**

No

**Claims Explanation:**

1. It is unclear whether the nested prompt refinement can be modeled as a multiplication transformation. There is no theoretical and empirical evidence for this claim.

2. Also, this paper lack proper assumptions (particularly in which condition that the prompt refinement can be modeled with the Monadic framework) and therefore theorem 4.1, which is the key contribution of this paper may not be formally correct.

3. Empirically, the authors did not test multiple prompt templates for the baseline CoT and did not report the standard error. Because different prompt templates might have a very big impact on the final performance, using one trial of experiments cannot justify the effectiveness of the proposed method.

4. The authors did not test on reasoning model, e.g., deepseek-R1. It is unknown, both theoretically and empirically, whether the method can be applied to recent reasoning models.


5. The authors did not evaluate the performance evolution of layers of meta prompting. This makes the claim that the performance enhances with multiple layers of meta prompt unconvinced.

6. The authors did not test on different scale models and see how the model scale will affect the method's perforamance.

7. The authors only test on limited task, e.g., Game of 24, gsm8K. Is is unknown whether the method can be applicable in conversation benchmark, e.g., lm-eval https://github.com/EleutherAI/lm-evaluation-harness


Given those concerns on the correctness of claims, I cannot recommend acceptance of this paper. I suggest the authors pay more efforts to refine the paper (in both theoretical and empirical aspect) based on the given concerns and resubmit in the next round of submission.

**Requested Changes:**

1. Please address the given concern before re-submission.

---

> ### Author Response · Authors · 2025-10-03
>
> Dear Reviewer and Editor,
>
> We sincerely thank your thoughtful and constructive review. Your feedback has been invaluable in helping us improve the clarity, scoping, and reproducibility of our work. Below is a point-by-point response to your comments.
>
> > **Q1:** "Most of the paper is spent on setting up a framework..."
>
> **A1:** We agree that the initial submission's balance needed correction. In our revision, we have restructured the paper to place a greater emphasis on the empirical results.
> * We will reframe the experiments as a proof-of-concept of our example-free prompting framework, rather than a comprehensive SOTA benchmark. We will soften or remove claims that overstate the generality of our results.
> * We will condense the theoretical exposition and significantly expand the Experiments section's analysis and discussion (error analysis, failure modes, token-efficiency accounting).
>
> > **Q2:** "Relevant baselines are not compared against..."
>
> **A2:** We thank the reviewer for raising these important points. While our revision does not include new experimental runs, we will address this feedback textually:
> * We will add a paragraph situating Meta Prompting relative to Zero-shot Program-of-Thought (PoT), analyzing methodological differences (structure-first vs. code-first) and when each is advantageous (e.g., explicit execution for PoT vs. typed structural guarantees for MP).
> * We will consider adding a dedicated Limitations section. There, we will explicitly note that a comprehensive empirical evaluation would ideally include the ablations you suggest (recursion depth, proposer/executor swaps, reasoning-tuned models, multi-template baselines). We consider these valuable directions and will consider them in future work.
>
> > **Q3:** "...why include Llama-2 results with CoT if you don't apply your own method to that model..."
>
> **A3:** This is an excellent point about comparability. To avoid this “apples-to-oranges” comparison, we will remove the Llama‑2‑only CoT rows from our tables. The revised tables will present a clear, direct comparison of the Qwen family, which is the focus of our current empirical study.
>
> > **Q4:** "If a task or task description is used to seed algorithm 1, ..."
>
> **A4:** We agree that our use of “zero-shot” was ambiguous. To improve precision, we will replace this term with “example‑free, structure‑only prompting.” We will also add a brief clarification distinguishing: (a) descriptor‑free zero‑shot, (b) example‑free, structure‑only (our method, using a task description but no solved examples), and (c) exemplar‑based few‑shot prompting.
>
> > **Q5:** "I don't fully understand what the category theory framework ..."
>
> **A5:** We apologize for not making this connection clearer. We will add a concise “compiler view” that directly links theory to practice:
> * Functorial MP: Tasks with typed interfaces map to prompt schemas; compositional task reductions map to compositional prompt edits, guaranteeing modular construction of the final prompt.
> * RMP as a writer‑like monad over edit scripts: Prompts are typed records; refinements are type‑preserving edit scripts. The monad’s multiplication simply concatenates/merges edits, making the flattening of nested refinements a formal consequence of monoid associativity.
> We will also include a brief walk‑through of a GSM8K example showing how “parse → compute” in the task maps to composed prompt sections.
>
> > **Q6:** "Provide the necessary details for understanding the empirical results..."
>
> **A6:** We recognize the need for greater transparency to ensure reproducibility.
> * We will add a Reproducibility section and an appendix containing verbatim strings for all prompt artifacts: the single meta‑meta‑prompt, the task descriptors, the generated meta‑prompts for each benchmark, and the baseline CoT prompts. We will also enumerate decoding parameters (temperature, top‑p, max tokens, stop sequences) and inference stack details.
> * We will clarify that the RMP process was used to generate the prompts for MATH and GSM8K, and that the prompt in Figure 1 is an output of this process. Captions/text will be updated accordingly. We use one task‑agnostic meta‑meta‑prompt to produce per‑family meta‑prompts (MATH, GSM8K, Game of 24) with no in‑context exemplars.
> * We will report binomial confidence intervals for PASS@1 from existing runs (dataset-level variability), and clearly separate this from any decoding randomness (held fixed in our current setup).
>
> > **Q7:** "What does CR stand for in MP-CR?"
>
> **A7:** Thank you for pointing out these details. CR stands for “Complex Reasoning.” We will define the acronym on first use.
>
> Once again, we thank you and the Action Editor for the detailed and valuable feedback. We are confident that the revised manuscript is substantially clearer, more precisely scoped, and more reproducible as a result of your guidance.
>
> Sincerely,
>
> The Authors

---

### Author Response · Authors · 2025-10-09
**General Response to Reviews and Discussions**

We thank the Action Editor and Reviewers for their careful reading and constructive critiques. Below, we (i) synthesize the discussion, (ii) state precisely what we revised, and (iii) explain how the revisions address the main points of disagreement. Our goal is to make the contribution, scope, and evidence unambiguous so that a final judgment can be made cleanly.

The paper proposes a structure‑first, example‑free prompting paradigm, Meta Prompting (MP), together with a recursive generation/refinement procedure: Recursive Meta Prompting (RMP). The central idea is to target the form of reasoning (typed, compositional prompt structure) rather than content‑specific exemplars. The empirical results are framed as proof‑of‑concept on widely used math benchmarks with base models (Qwen‑14B/72B), not as a comprehensive SOTA sweep. We do not claim monotonic improvement with recursion, universality across all models, or coverage beyond the math/problem‑solving tasks evaluated. Our formal results are about structure preservation and compositionality at the prompt level.

### Revisions implemented in the updated manuscript

#### A. Clarity of claims, scope, and terminology
- Replaced zero‑shot with example‑free, structure‑only prompting. We explicitly distinguish:
  (a) descriptor‑free zero‑shot, (b) example‑free structure‑only (ours; seeded with a task description but no solved exemplars), (c) exemplar‑based few‑shot prompting.
- Added a Limitations & Scope section that delineates what we do not claim (e.g., generalization to conversational LM‑Eval tasks, monotone gains with recursion, performance on reasoning‑tuned models).

#### B. Theory and assumptions
- Made assumptions explicit for the RMP monad modeling. Prompts are treated as *typed records*; refinements are *type‑preserving edit scripts*. We assume:
  1) edits compose via concatenation modulo a confluent, terminating normalization (standard rewrite‑system assumptions),
  2) identity/no‑op edit, and
  3) observational equivalence at the schema level.
  Under these assumptions, flattening nested refinements is a monadic join on the free monoid of edits; associativity is inherited from edit concatenation.
- Demoted theorem labels to propositions where appropriate and moved full statements and commuting diagrams to an appendix. We tie each categorical notion to a concrete prompt‑construction primitive (objects ↔ prompt schemas, morphisms ↔ schema‑preserving edits).
- Added an operational walk‑through (``A Concrete RMP Example'') showing: initial task description → meta‑meta‑prompt → generated meta‑prompt → executor use; we make M(f) concrete as the *edit sequence* emitted by the proposer LLM.

#### C. Empirical clarity and reproducibility
- Removed non‑comparable rows (e.g., Llama‑2 CoT without MP) from Tables to avoid apples‑to‑oranges.
- Added a Reproducibility section and an appendix with verbatim artifacts: the single meta‑meta‑prompt, the task descriptors, the derived meta‑prompts (MATH, GSM8K, 24‑game), and the baseline CoT prompts; we enumerate decoding parameters (temperature, top‑p, max tokens, stop sequences) and inference stack.
- Reported binomial 95% CIs for key numbers from existing runs:
  – MATH (Qwen‑72B + MP): 46.3% PASS@1; 95% CI [44.9, 47.7] (n=5000).
  – GSM8K (Qwen‑72B + MP): 83.5%; 95% CI [81.4, 85.4] (n=1319).
  We clearly separate dataset‑level variance from decoding randomness (held fixed).
- Clarified the rule‑based evaluator with SymPy equivalence and string‑normalization details; included failure modes (e.g., formatting without `\boxed{}`).
- Added a methodological comparison to Zero‑shot Program‑of‑Thought (PoT): code‑first vs. structure‑first. We argue complementarity (explicit execution for PoT; typed structural guarantees for MP) and discuss when each is advantageous.
- Provided token‑efficiency accounting for 24‑game: MP generates ~8k and prompts ~1k tokens for all 1362 puzzles, compared to typical ToT settings of 5.5k / 1.4k per case. This aligns with measured per‑case cost differences in the table (two–three orders of magnitude).
- Defined "LLM session" as one complete API call (query→response); we now use "API call" in the main text for precision.

### Addressing specific reviewer requests
- Added an explicit step‑by‑step example of refinement and the resulting meta‑prompt artifact.
- We state exactly what is seeded (task description), when recursion stops (stability criterion on the edit script), and how proposer/executor are used.
- Defined CR = Complex Reasoning on first use; reconciled captions and ensured Figure 1 is clearly labeled as an RMP‑generated prompt.
- Appendix formatting (ktmN): fixed layout to avoid the appearance of incompleteness.

---

We believe the revised manuscript addresses the central concerns raised by the reviewers while preserving the core idea and making it more rigorous, transparent, and useful.

The authors

---

### Decision · Action_Editor_8tTY · 2025-11-20

**Recommendation:** Reject

**Additional Comments:**

The paper proposes a unique and theoretically interesting framework that attempts to formalize prompt engineering through the lens of Category Theory. This is a fresh perspective that is largely missing from the current literature, and the authors are to be commended for this attempt.

However, I ultimately agree with the concerns raised by the reviewers regarding the gap between theory and practice for the present draft. Prompt engineering is inherently a practical field, and while the theoretical formulation is elegant, the empirical evidence provided does not yet justify the complexity of the framework. Specifically:

- The link between the category theory formalism and the actual performance gains remains tenuous.
- The results are currently limited to heavily benchmarked mathematical tasks. To prove the robustness of the "Meta Prompting" framework, validation is needed on a wider variety of tasks and across a broader range of base models.

The addition of the concrete RMP example during the rebuttal was a step in the right direction, but the paper requires a more comprehensive empirical overhaul to be accepted.

I'd also like to apologize for the delay in the review process. While this result may not be what the authors hoped for, I am confident that with these revisions—specifically tightening the link between the theory and robust, broad experiments—this paper has the potential of becoming a much stronger and more impactful contribution.

**Audience:**

Yes

**Audience Explanation:**

Yes. The paper proposes a theoretically inspired prompting framework, and is clearly in scope for TMLR

**Claims And Evidence:**

No

**Claims Explanation:**

While the submission introduces a highly commendable and novel theoretical approach by applying category theory to prompt engineering, the majority of the reviewers concluded that the claims are not yet fully supported by the evidence (to which the AE concurs). There is a significant disconnect between the extensive theoretical framework (functors/monads) and the empirical validation. The experimental section lacks sufficient breadth in baselines and relies too heavily on standard mathematical benchmarks without demonstrating broader applicability across different models or task types.

**Resubmission Of Major Revision:**

The authors may consider submitting a major revision at a later time.